# Association between Problematic Internet and Mobile Phone Use, autistic traits, and psychological distress among adults: A cross-sectional survey

Matilda Floris[1,2]\*, Claudio Gentili[1,2]

**1** Padova Neuroscience Center (PNC), University of Padova, Padova, Italy, **2** Department of General Psychology, University of Padova, Padova, Italy

\* matilda.floris@phd.unipd.it

## Abstract

Problematic Internet Use (PIU) and Problematic Mobile Phone Use (PMPU) are emerging public health concerns associated with various mental health conditions, including autistic traits. However, PIU and PMPU remain poorly understood, and the role of psychological distress as a contributing factor has not been fully clarified. This study examined frequency and associations of autistic traits, PIU and PMPU, and substance use in adults, accounting for psychological distress. An online cross-sectional survey was conducted among 420 Italian adults aged 18 – 65, to assess autistic traits (Autism Spectrum Quotient, AQ), PIU (Uso e Abuso di Internet-2, UADI-2), PMPU (Mobile Phone Problematic Use Scale for Adults, MPPUS), substance use (Alcohol, Smoking and Substance Involvement Screening Test, ASSIST), and psychological distress (Kessler Psychological Distress Scale, K10). Correlation analyses, MANCOVA and Multivariate Multiple Regression (MMR) were performed to examine associations between variables, controlling for psychological distress. Young adults (18–24 years) showed higher levels of Internet (M = 66.36, SD = 13.56, 95% CI [63.87, 68.85]) and mobile phone use (M = 61.41, SD = 14.25, 95% CI [58.92, 63.9]). Autistic traits correlated positively with both PIU and PMPU, while psychological distress was positively associated with all outcomes, including substance use in younger participants. MANCOVA indicated significant effects of autistic traits and psychological distress on outcomes. MMR showed that higher autistic traits predicted higher PIU and PMPU, while psychological distress additionally predicted substance use. Autistic traits and psychological distress were significantly associated with PIU and PMPU, with psychological distress showing stronger associations. PIU and PMPU may be specifically linked to autistic traits, whereas substance use may be more related to psychological distress, suggesting different pathways. Further efforts should prioritize interventions targeting distress and emotion-regulation skills, particularly among young adults and individuals exhibiting PIU and PMPU.

**Data availability statement:** The study materials are publicly available on the OSF repository:https://osf.io/yvrfq/overview. The repository includes the anonymized dataset ("dataset.xlsx"), the codebook describing all variables ("codebook.xslx"), and the complete R Markdown script ("script.Rmd") containing all statistical analyses performed in this study.

**Funding:** This work was supported by the National Recovery and Resilience Plan (PNRR) of the European Union (NextGeneration EU) (39-412-19-DOT18G59FB-4181 to M.F.). The funders had no role in the study design, data collection and analysis, decision to publish, or preparation of the manuscript.

**Competing interests:** The authors have declared that no competing interests exist.

## Introduction

The use of Internet and mobile phone has become essential in daily life worldwide, with a sharp increase in recent decades. According to the Censis report on media use in Italy (2025), 90.1% of the population use Internet, 89.3% use smartphones, and 85.3% social networks [1]. Whereas Internet provides benefits, such as the acquisition of knowledge, social connection, and entertainment, its excessive use has been associated with negative outcomes, including exposure to inappropriate content, cyberbullying, distraction from school or work, social withdrawal, as well as physical health (e.g., headaches and neck pain) [2] and mental health concerns (e.g., depression, anxiety, substance use, psychological distress) [3–6].

In literature, the term "Internet addiction" generally refers to an excessive and maladaptive use of Internet that interferes with daily functioning. Originally, it was conceptualized in analogy to substance use disorder (SUD) and gambling disorder due to shared features – loss of control, mood changes when offline, dominance of thought, withdrawal symptoms, and tolerance – [7], while some other authors have instead suggested classifying it as an impulse control disorder [8]. To date, the classification of "Internet addiction" remains debated, with some authors questioning whether it should be considered a distinct disorder or rather a comorbid manifestation of other mental conditions [7]. Only problematic Internet use related to gaming is officially recognized as a primary disorder – namely, Gaming Disorder in the International Classification of Diseases 11th Revision (ICD-11), and Internet Gaming Disorder as a condition for further study in Diagnostic and Statistical Manual of Mental Disorders 5th Edition, Text Revision (DSM-5-TR) [9]. Due to this lack of consensus, the term Problematic Internet Use (PIU) was preferred in this study to describe a preoccupation with and lack of control over diverse Internet-related behaviours – including general excessive use, gaming, smartphone use, and social networking – that lead to clinically significant distress or impairment in multiple areas of functioning. Heterogeneity in terminology and criteria makes it challenging to estimate the prevalence of PIU. A recent meta-analysis of 113 epidemiological studies across 31 countries reported a global prevalence of approximately 7% from 1996 to 2018 [10]. In Italy, a study among high school students found high levels of PIU in 14.2% of boys and in 10.1% of girls [11].

Similarly, the excessive use of smartphone has recently received growing attention due to its portability and its ability to provide constant and immediate access to the Internet and engagement in a wide range of online activities. This behaviour has been described as "mobile phone addiction" or "smartphone addiction", emphasizing its addiction features similar to those of PIU. However, as for PIU, the term Problematic Mobile Phone Use (PMPU) was considered more appropriate to describe difficulties in controlling time spent on mobile phones, which can negatively impact daily functioning. The current debate concerns whether PMPU can be considered a subcategory of PIU or as an independent phenomenon. PMPU and PIU appear to overlap to some extent, as several studies have found positive correlation between them [12–15]. Indeed, both refer to similar behavioural pattern, considering that individuals are almost always connected to the Internet while using a mobile phone. Despite

these considerations, some studies have identified the differences between the two constructs. For example, a recent net-work analysis on 4070 Slovak adolescents found that PMPU and PIU differ at psychological level, revealing gender and risk factor differences [12]. Specifically, externalizing and internalizing problems, fear of missing out (FoMO) and hope-lessness were strongly associated with PIU, except for FoMO which showed a strong correlation with PMPU. Moreover, externalizing problems were central in the boys' network, whereas internalizing, externalizing problems and resilience were central in the girls' network. A systematic review of 117 studies among university students also found distinct risk factors predicting Internet overuses (including PMPU and IGD). While negative affectivity was a common predictor, FoMO, type of use, duration and frequency of mobile phone use were specific to PMPU [16]. Several studies have also confirmed gender differences, with girls being more susceptible to PMPU than boys [12,17]. Furthermore, a meta-analysis involving 140776 participants from 94 independent samples worldwide found that women were more likely to exhibit social media addiction, while men were more likely to show IGD [18]. Taking into account this evidence and the pervasive role of mobile phone in daily life (e.g., for social media use such as WhatsApp and Instagram), assessing mobile phone use appears essential when investigating Internet-related behaviours.

### PIU, PMPU, autistic traits and psychological distress

Internet-related behaviours, such as PIU and PMPU, have been linked to several adverse consequences. For instance, PMPU has been linked to social anxiety [19], poor communication skills [20], reduced sleep quality [21] and difficulties in emotion regulation [22]. Moreover, a systematic review of 23 studies reported consistent associations between PMPU and anxiety, depression and chronic stress [3]. In an Italian sample of young adults, psychological distress and the nocturnal use of smartphone to attend social networks were associated with a higher risk of PMPU [23]. Furthermore, although life stressors emerged as the strongest predictor of depression severity, PIU, PMPU and problematic social media use accounted for approximately 10% of the variance, suggesting that problematic online behaviours significantly contributed in the development of depressive symptoms among university students [24].

Similarly, PIU has been increasingly linked to depression [25,26], social anxiety [27,28], attention-deficit and hyper-activity disorder (ADHD) [29,30] and autistic spectrum disorder (ASD). In addition, a recent meta-analysis of Muris et al. (2025), including 46 studies and 42274 participants, found a significant association between ASD or high autistic traits and PIU, excessive gaming, and PMPU (r = 0.26) [31]. Consistently, the systematic review of Murray et al. (2022) showed positive association between PIU and subclinical autistic traits with weak to moderate effect sizes [32]. In Italy, Sulla et al. (2023) investigated subclinical autistic traits and PIU among 141 university students, finding that autistic traits moderated the link between quality of life and PIU [33].

Regarding PMPU, social autistic traits were positively associated with this behaviour, with anxiety and executive dysfunction mediating their relationship in a sample of high school students [34]. Similarly, a study among 1103 Chinese college students found that social anxiety and loneliness mediated the association between autistic traits and PMPU [35].

Overall, scientific evidence suggests that both PIU and PMPU are linked to psychological distress (e.g., anxiety, depression) and autistic traits, although the interplay between these factors remains largely unexplored.

### PIU, PMPU, autistic traits and substance abuse

The association between substance use and Internet-related behaviours has been investigated in recent studies. Evi-dence from a large sample of US college students showed that individuals with higher level of PIU were at significantly greater risk of engaging in substance misuse behaviours [39]. This finding was confirmed by a large-scale study among Korean adolescents, in which students at higher risk of PIU were associated with greater consumption of alcohol, tobacco and other drugs [40]. Additionally, problematic alcohol use has been associated to PIU, IGD and problematic use of social media among adolescents [41] and college students [42]. Tobacco and alcohol use have also shown significant associa-tion with both PIU [43] and PMPU [44,45].

Nevertheless, the risk of substance abuse within the autistic population remains less clear, and studies involving individuals with elevated autistic traits are lacking. A systematic review reported that the prevalence of substance abuse among individuals with ASD ranged from 1.3% to 36%, due to variability in diagnostic measures and sample characteristics [46]. A survey of 2386 participants revealed that autistic individuals were less likely than non-autistic individuals to consume alcohol regularly, binge drink, smoke, or use drugs [47]. Conversely, a recent systematic review found that ASD population is significantly more susceptible to substance use and related disorders [48]. Interestingly, one study found that 20% of treatment-seeking SUD patients showed elevated autistic traits [49].

Taken together, these findings highlight inconsistent and understudied field, suggesting the need to clarify the relationship between PIU and PMPU and substance misuse both in population with and without high autistic traits.

## The study

In the present study, PIU and PMPU were investigated in association with autistic traits and substance abuse, while controlling for psychological distress. Autistic traits may be associated with PIU and PMPU due to (a) deficit in social interactions and communication, and (b) restricted interests and repetitive behaviours (RIRBs), such as gaming [31,50]. Internet may serve as an alternative to face-to-face interactions, offering a less demanding social environment, as well as a space to engage in information-seeking behaviours that support their RIRBs. Moreover, higher level of psychological distress (e.g., anxiety, depression) may also intensify social difficulties and increase reliance on Internet use as coping a strategy, along with its established association with PIU, PMPU and substance use. In addition, individuals with elevated autistic traits may use substances as self-medication behaviour to manage mental health symptoms [47].

This interplay between autistic traits and psychological distress in the association with PIU and PMPU can be interpreted within the Interaction of Person-Affect-Cognition-Execution (I-PACE) model, which explains the development and maintenance of Internet-related disorders through personal, affective, cognitive and executive factors [51,52]. According to I-PACE model, PIU results from the interaction between predisposing (i.e., personality, social cognition, neurobiological, psychopathology, biopsychological constitution and motives of Internet use), moderating (i.e., coping style and Internet-related cognitive biases) and mediating factors (e.g., affective and cognitive responses to situational triggers) that impair executive functioning.

A further aim of this study was to assess frequency of PIU, PMPU, psychological distress, and substance use among adult population. In fact, considering that most of the research has focused on adolescents and young adults, this study explores potential differences across adult age groups in Internet-related behaviours, substance use and psychological distress. Specifically, the research questions were:

a) What is the frequency of PIU, PMPU, psychological distress and substance use among adults, and this frequency differ across age groups?

b) Do participants with different autistic traits show significant differences in PIU, PMPU and substance use after controlling for psychological distress?

## Methods

### Pre-registration

The study was pre-registered on the Open Science Framework (OSF): https://osf.io/d2nbg/overview?view_only=17adeb-498f104e24976c4bf34948256a. One deviation from the pre-registration occurred. Specifically, the pre-registered sample size (n = 500), which was intended to allow the formation of five age groups with 100 participants each (18 − 25; 26 − 35; 36 − 45; 46 − 55; 56 − 65), was not reached within the predefined data collection period (i.e., March 2025, as specified in the pre-registration). Therefore, participants were divided into four age groups based on quartile distribution of the

recruited sample (see the next paragraph and Fig 1 for details). This decision was made to ensure balanced group sizes that better reflect age distribution of the recruited sample. However, the final sample size was sufficient for the overall analyses. As specified in the pre-registration, sample size was determined using an online calculator (http://www.raosoft.com/samplesize.html), assuming a 95% confidence interval, 5% of marginal error, a response distribution of 50%, and a population size of 20000, which lead to a minimum required sample of 377 participants.

## Participants

The sampling followed a combination of convenience and snowball methods. Recruitment was carried out from 25/06/2024 to 31/03/2025 through advertisements on social media platforms (i.e., Facebook, Instagram, WhatsApp), word of mouth, mailing list and flyers at University of Padova and at the Mental Health Service for university students. Inclusion criteria were: (a) ≥ 18 years old; (b) good knowledge of Italian (native speaker); (c) absence of intellectual disability or cognitive impairment that could compromise the ability to provide informed consent. These criteria were assessed through three self-report questions at the beginning of the survey.

Of the 655 individuals who started the survey, 61 participants were excluded for not meeting the inclusion criteria, and 164 were excluded due to duplicate or incomplete responses. The final sample consisted of 420 participants, corresponding to a completion rate of 70.7% (420/594). Flowchart and details on exclusion criteria are provided in Fig 1. Participants were divided into 4 age groups based on quartile distribution: (a) 18 − 24 (n = 114); (b) 25 − 36 (n = 107); (c) 37 − 49 (n = 106); 50 − 65 (n = 93). The study was approved by the Institutional Review Board of University of Padova (code: 583-b) and reported according to the STROBE checklist for cross-sectional studies [53].

## Survey

A cross-sectional design was employed using an open online survey. Participation was voluntary, anonymous, and no reimbursement was provided. The survey included the Italian adoption of the following questionnaires (see next sections for a brief description):

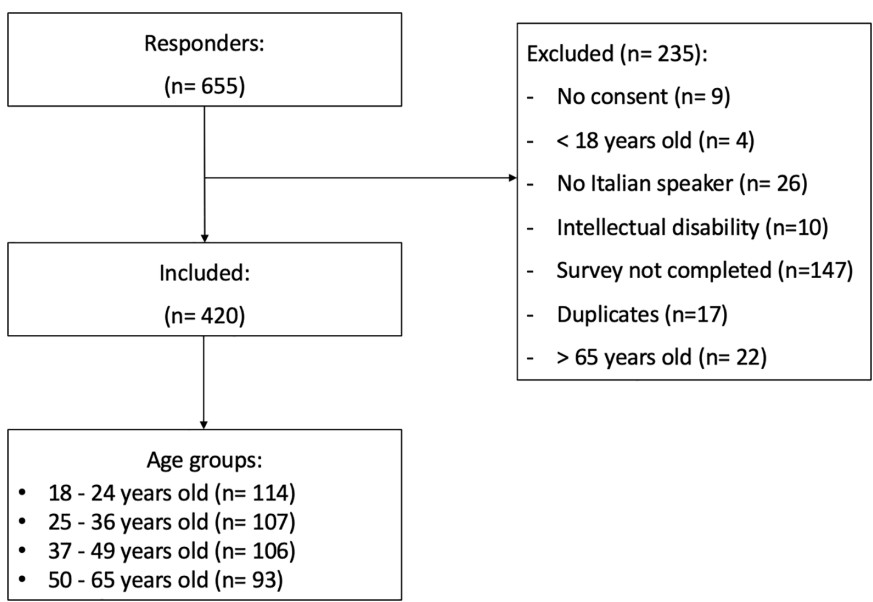

**Fig 1. Flowchart of participants of the study.**

1. Kessler Psychological Distress Scale (K10) [54] to assess psychological distress

2. Autism Spectrum Quotient (AQ) [55] to measure autistic traits

3. Uso e Abuso di Internet-2 (UADI-2; *Internet Use-Abuse and Addiction*) [56] for PIU

4. Mobile Phone Problematic Use Scale for Adults (MPPUS) [57] for PMPU

5. Alcohol, Smoking and Substance Involvement Screening Test (ASSIST) [58] to assess substance intake

6. Sociodemographic questionnaire, developed ad hoc

A subsample of participants (n = 230) also completed the Internet Addiction Test (IAT) [59] to assess PIU, as this measure was added later to the survey to enhance comparability with the international literature. This subsample will be analysed separately (manuscript in preparation). Participants who completed the IAT did not differ from those who did not complete it with regard to sociodemographic and mobile phone use variables (see Supporting Information S1 Table).

All questionnaires were presented in a fixed, non-randomized order, following the sequence listed above. The informed consent appeared immediately after clicking the survey link, and participants could accept and proceed or refuse and exit. The informed consent described the study aims, instruments, investigator details, data management and length of compilation. The survey was powered by Qualtrics, which generate a standard link for the survey. All items of each instrument were presented on the same page, while the sociodemographic questionnaire was divided into sets of 2 – 3 items per page. The survey consisted of 21 – 28 pages (depending on ASSIST compilation), and took approximately 20 minutes to complete. All items were mandatory, although participants can interrupt the completion at any time. A back button was not inserted in the survey to prevent answer modifications. To ensure data quality, a control item was included (*This is a control question. Please select B to demonstrate your attention*) to identify inattentive or random responses. Qualtrics automatically recorded metadata, such as date and time of completion, progress percentage, IP addresses, response ID (unique and random), and duplicate responses. Response IDs and IP addresses were never used to the identify participants. Duplicate responses were flagged by Qualtrics using cookies in participants browser, and were deleted during data analysis. Participants were allowed to leave and re-enter the survey to finish it later. The survey was tested before dissemination with colleagues, to check its usability, functionality, and the presence of typographical errors. They did not re-complete the survey, and their responses were excluded from the data analysis. Access to the online survey editor, results and the dataset was protected by password.

### Measures

**Independent variables.** Autistic traits were considered the main independent variable. In exploratory analysis, sociodemographic variables were also included as factors.

Autistic traits were assessed with the Autism Spectrum Quotient (AQ) [55,60], a self-report questionnaire tailored to identify them in adults. The AQ consists of 50 items, divided into 5 domains (10 items each): social skills, communication, imagination, attention to detail, and attention switching. Responses are given on a four-point scale: "definitely agree", "slightly agree", "slightly disagree", and "definitely disagree". Total scores range from 0 to 50, with higher scores indicate higher autistic traits. A cut-off of 26, adopted in this study, correctly identified 83% of the individuals with traits, while a cut-off of 32 is used in clinical sample [61]. The AQ in this study shows a global Cronbach's α of.82, indicating a good internal consistency. Across domains α values showed poor to acceptable internal consistency (communication: α = .66; social: α = .72; imagination: α = .37; attention details = .67; attention switching = .62) [55]. Given the poor internal consistency of the Imagination subscale, the main analyses were re-run excluding this subscale. The results of these analyses were consistent with the original findings; therefore, they are reported only in Supporting Information S2 Table. The AQ has been validated in an Italian sample [60].

In exploratory analysis, sociodemographic variables, assessed through self-report items with multiple-choice answers, were included in the models. Details of sociodemographic variables – gender, age groups, education level, geographic origin, marital status, housing condition, occupation, economic level, presence of chronic disease, presence or absence of psychological disorders, issues with justice (none, civil, penal), traumatic event, time spent on social network, time spent on mobile phone, type of smartphone use – are provided in Supporting Information S3 Table.

**Dependent variables.**  PIU, PMPU, and substance use were the dependent variables of the study.

The Uso-Abuso e Abuso di Internet 2 (UADI-2; *Internet Use-Abuse and Addiction*) is a 24-item self-report questionnaire that assess Internet use and misuse [56]. Responses are rated on a five-point Likert scale (1 = "absolutely false for me", 5 = "absolutely true for me"). The UADI-2 measures 4 dimensions: dissociation, impact on real life, addiction symptoms, identity and sexuality. Total scores range from 24 to 120, with cut-offs that indicate normal use (≤ 62), Internet misuse (63 – 74) and Internet addiction (> 74). In the current study, the Cronbach's α showed good internal consistency (α = .89) for global score, and from poor to good for each dimension (dissociation: α = .76, impact on real life: α = .49, addiction symptoms: α = .81, identity and sexuality α = .75).

The Mobile Phone Problem Usage Scale (MPPUS) is a self-report instrument aimed to assess PMPU in general population. The original scale includes 27 items covering presence of tolerance, craving, negative consequences in daily life, and loss of control [57]. For the purpose of this study, the Italian validation was used, consisting of 24 items as items 4, 7, and 17 were excluded due to unclear factor loading [62]. Items are rated with five-point Likert scale (1 = "not all true", 5 = "very true"). Total scores range from 24 to 120, but a recognized clinical cut-off is not available. Following previous studies (e.g., [63]), the tertile distribution were used to categorized participants into low (scores 24 – 46), medium (47 – 60) and high (> 60) smartphone use. Given the sample-dependence of the tertiles, they were used exclusively for descriptive purpose, whereas the MPPUS score was treated as continuous variable in all main analyses. The Italian version presents a two-factor model, the "Withdrawal and social aspects" and the "Craving and escape from other problems". Internal consistency in this study was excellent (Cronbach's α = .91) for global score, good for Craving and escape from other problems (Cronbach's α = .84), and acceptable for Withdrawal and social aspects (Cronbach's α = .78).

The Alcohol, Smoking and Substance Involvement Screening Test V3.0 (ASSIST) was developed by the WHO to screen lifetime use of tobacco, alcohol, cannabis, cocaine, amphetamine-type stimulants (including ecstasy), inhalants, sedatives, hallucinogens, opioids and other drugs [58]. ASSIST comprises 8 questions, producing a scores that classify risk level for each substance as low (scores < 3), moderate (4 – 26), or high (≥ 27) [58]. Alcohol scores were slightly different, with a low risk defined by scores < 10, moderate 11 – 26 and high ≥ 27. Cronbach's α values ranged from .73 (tobacco) to .92 (alcohol), in the validation paper [58]. Cronbach's α was not calculated for this questionnaire due to lack of variability in responses across different substances. The ASSIST has been widely used in several countries and cultures [64–66].

**Covariate.**  Psychological distress was included as a covariate in the models. It was measured using the Kessler Psychological Distress Scale (K10), a 10-item self-report questionnaire to screen the level of distress experienced in the past 30 days [54]. Items regard, for instance, symptoms of depression, anxiety, tidiness and restlessness. Items are rated using a five-point scale (1 = "none of the time", 5 = "all of the time"), leading to a total score ranging between 10 and 50. Scores and can be interpret as follow: no distress (< 20), mild distress (20 – 24), moderate distress (25 – 29), severe distress (> 30). The internal consistency showed a Cronbach's α of .92 in this study.

## Data analysis

The first research question was addressed using descriptive statistics. Descriptive statistics were performed to examine sample characteristics, and estimate the frequency of autistic traits (AQ), PIU (UADI-2), PMPU (MPPUS), and substances use (ASSIST), in the overall sample and across age groups. Specifically, means and standard deviations were calculated

for continuous variables, and percentage for categorical variables. Differences across age groups were investigated using the Kruskal-Wallis test for continuous variables, and Pearson's Chi-squared test or the Fisher's exact test for categorical variables. Fisher's Exact Test was applied when expected frequencies were below 5; otherwise, chi-squared tests were used. Ninety-five percent confidence intervals (CIs) were also reported for the key results.

The second research question was examined through correlational analyses, MANCOVA and follow-up ANCOVAs. Correlational analyses were conducted within age groups to assess (a) the association between the autistic traits and each dependent variable (UADI-2, MPPUS, and ASSIST scores), and (b) the association between psychological distress (K10) and the dependent variables to further explore its role as covariate. Analyses focused on tobacco and alcohol use, as only a small proportion of participants reported the use of other substances. Spearman's rho coefficient was used, since the variables were not normally distributed (Shapiro-Wilk normality test: $p < .001$), and was interpreted according to literature [67]. Ninety-five percent confidence intervals (CIs) were also reported for the correlational analyses.

Finally, a one-way between-subjects Multivariate Analysis of Covariance (MANCOVA) was used to examine differences between independent variable (autistic traits) and dependent variables (UADI-2, MPPUS, and ASSIST scores), controlling for covariate (K10 score). Autistic traits were treated as categorical (low autistic traits vs high autistic traits), while psychological distress (K10) and dependent variables (UADI-2, MPPUS, and ASSIST scores) as continuous variable. A sensitivity MANCOVA was also conducted between autistic traits and dependent variables with age and K10 score included as continuous covariates. This analysis was performed to address the deviation from pre-registration and ensure that the observed effects remained robust regardless of age grouping.

The pre-registered MANCOVA model revealed two methodological issues: (i) a substantial imbalance between individuals with low autistic traits (n = 359, 85%) and with high autistic traits (n = 61, 15%) and (ii) the zero-inflated distribution of ASSIST-tobacco (51%, n = 216 non-users) and ASSIST-alcohol (21%, n = 90 non-users). To address the imbalance between autistic traits, a Multivariate Multiple Regression (MMR) model was deemed more appropriate, as it allow to use AQ as a continuous factor. The MMR was conducted with AQ and K10 as continuous predictors, testing multiple regression models for each dependent variable simultaneously. To account for zero-inflated nature of the ASSIST-tobacco and -alcohol, follow-up Hurdle Models analyses were performed. These models separately estimate the probability of tobacco and alcohol use (0 vs > 0) and the intensity of use among users. Accordingly, in the present study, we report the pre-registered MANCOVA, alongside with the MMR model and the Hurdle models analyses to evaluate the robustness of the findings under alternative and methodologically appropriate analyses

Exploratory analyses were conducted using one-way between-subjects analysis of covariance (ANCOVA) to further investigate the association between dependent variables (UADI-2, MPPUS, and ASSIST scores), independent variable (autistic traits) and the covariate (K10), adjusting for sociodemographic variables. Post-hoc corrections for multiple comparisons with Tukey's tests were applied when ANCOVAs factors with more than two levels were significant.

Additionally, Spearman's correlations and MMR analyses were re-run as sensitivity analyses excluding the Imagination subscale of the AQ due to its poor internal consistency. Additional descriptive analyses, MANOVA and ANOVAs exploring the difference between males and females were reported in Supporting Information S4 Table and S5 Table.

The standard $p < .05$ was set as significance threshold. All the analyses were performed in R Studio 4.4.2 [68] using *dplyr* [69], *gtsummary* [70] and *emmeans* [71] packages with MacBook Pro M3.

## Results

### Descriptive characteristics

Table 1 reports selected characteristics of the included participants (n = 420, 334 females) across age groups. The mean age of the sample was 38.25 (SD = 13.64). Of these 420 participants, 418 (99%) were Italian, 263 (63%) were originally from Islands, 165 (39%) had a high school education level, 132 (31%) were married, and 263 (63%) were employed. Significant differences emerged among age groups in education level, geographic origin, marital status, housing situation,

**Table 1. Selected characteristics of the included participants (n = 420).**

| Variables | Age groups | | | | | Statistical test | p-value |
|---|---|---|---|---|---|---|---|
| | Overall N = 420 | 18–24 n = 114 | 25–36 n = 107 | 37–49 n = 106 | 50–65 n = 93 | | |
| **Gender** | | | | | | Fisher's Exact Test | 0.366 |
| Female | 334 (80%) | 89 (78%) | 82 (77%) | 84 (79%) | 79 (85%) | | |
| Male | 80 (19%) | 24 (21%) | 22 (21%) | 21 (20%) | 13 (14%) | | |
| Non binary | 2 (1%) | 1 (1%) | 0 (0%) | 1 (1%) | 0 (0%) | | |
| Prefer to not answer | 4 (1%) | 0 (0%) | 3 (3%) | 0 (0%) | 1 (1%) | | |
| **Education Level** | | | | | | Fisher's Exact Test | <0.001 |
| High school | 165 (39%) | 47 (41%) | 25 (23%) | 45 (42%) | 48 (52%) | | |
| Master's degree | 96 (23%) | 15 (13%) | 33 (31%) | 27 (25%) | 21 (23%) | | |
| Bachelor's degree | 89 (21%) | 47 (41%) | 25 (23%) | 9 (8.5%) | 8 (8.6%) | | |
| Post-lauream | 50 (12%) | 2 (2%) | 20 (19%) | 19 (18%) | 9 (9.5%) | | |
| Middle school | 20 (5%) | 3 (3%) | 4 (4%) | 6 (5.5%) | 7 (7.5%) | | |
| **Citizenship** | | | | | | Fisher's Exact Test | 0.249 |
| Italian | 418 (99%) | 112 (99%) | 107 (100%) | 106 (100%) | 93 (100%) | | |
| European | 2 (1%) | 2 (1%) | 0 (0%) | 0 (0%) | 0 (0%) | | |
| **Origin** | | | | | | Fisher's Exact Test | <0.001 |
| Islands | 263 (63%) | 36 (32%) | 63 (59%) | 91 (86%) | 73 (78%) | | |
| North Italy | 104 (25%) | 51 (45%) | 27 (25%) | 12 (11%) | 14 (15%) | | |
| Central Italy | 26 (6.6%) | 16 (14%) | 8 (7.5%) | 0 (0%) | 2 (2.4%) | | |
| Europe | 5 (1.2%) | 2 (2%) | 0 (0%) | 0 (0%) | 3 (2.6%) | | |
| South Italy | 21 (5.0%) | 9 (7%) | 8 (7.5%) | 3 (3%) | 1 (1%) | | |
| Extra-Europe | 1 (0.2%) | 0 (0%) | 1 (1%) | 0 (0%) | 0 (0%) | | |
| **Marital status** | | | | | | Fisher's Exact Test | <0.001 |
| Relationship | 171 (41%) | 68 (60%) | 67 (63%) | 27 (25%) | 9 (10%) | | |
| Married | 132 (31%) | 0 (0%) | 17 (16%) | 55 (52%) | 60 (65%) | | |
| Single | 95 (23%) | 46 (40%) | 23 (21%) | 14 (13%) | 12 (13%) | | |
| Divorced | 9 (2%) | 0 (0%) | 0 (0%) | 3 (2.8%) | 6 (6.5%) | | |
| Separated | 11 (2.5%) | 0 (0%) | 0 (0%) | 7 (6.2%) | 4 (4.3%) | | |
| Widowed | 2 (0.5%) | 0 (0%) | 0 (0%) | 0 (0%) | 2 (2.2%) | | |
| **Housing** | | | | | | Fisher's Exact Test | <0.001 |
| With partner | 200 (48%) | 9 (8%) | 54 (50%) | 70 (66%) | 67 (72%) | | |
| With parents | 76 (18%) | 52 (46%) | 17 (16%) | 4 (4%) | 3 (3.5%) | | |
| Alone | 65 (15%) | 12 (11%) | 20 (19%) | 17 (16%) | 16 (17%) | | |
| With roommates | 51 (12%) | 41 (36%) | 10 (10%) | 0 (0%) | 0 (0%) | | |
| Other | 27 (6.8%) | 0 (0%) | 5 (5%) | 15 (14%) | 7 (7.5%) | | |
| University residence | 1 (0.2%) | 0 (0%) | 1 (1%) | 0 (0%) | 0 (0%) | | |
| **Occupation** | | | | | | Fisher's Exact Test | <0.001 |
| Worker | 263 (63%) | 15 (13%) | 75 (70%) | 95 (90%) | 78 (85%) | | |
| Student | 81 (19%) | 72 (63%) | 9 (8.4%) | 0 (0%) | 0 (0%) | | |
| Student and worker | 48 (11%) | 27 (24%) | 19 (18%) | 1 (0.9%) | 1 (1.1%) | | |
| Unemployed | 16 (3.4%) | 0 (0%) | 4 (4%) | 9 (8.5%) | 3 (4%) | | |
| Retired | 10 (2.4%) | 0 (0%) | 0 (0%) | 0 (0%) | 10 (11%) | | |
| Unable to work | 1 (0.2%) | 0 (0%) | 0 (0%) | 1 (0.5%) | 0 (0%) | | |

and occupation (Fisher's Exact Tests, all *p* < .001). Other sociodemographic variables (i.e., economic level, psychological diagnosis, issues with justice, familiarity, and exposure to traumatic events), as well as psychological diagnosis details, were provided in Supporting Information S6 Table and S7 Table. No significant differences were found between age groups in gender (*p* = .36) and citizenship (*p* = .24). Survey completion time ranged from 5 minutes to 3 days, with a median value of 22 minutes. This variability was mainly due to ASSIST compilation and to the possibility to exit the survey and finish it later.

Mean values, standard deviations and Fisher's Exact Tests of mobile phone variables (time spent on mobile phone, time spent on social network, type of mobile phone use) were reported in Supporting Information S8 Table. Briefly, most participants reported using their mobile phone for 2 – 5 h/day (58%, n = 242), mainly for communications (49%, n = 206) and social networking (34%, n = 141), with an average daily social media use of 2 hours (62%, n = 261). Significant differences across age groups were observed for all variables (Fisher's Exact Tests, all *p* < .001).

Table 2 reports percentages or mean values and standard deviations, and statistical tests (Kruskal-Wallis tests for numeric variables and Chi-squared test) of autistic traits (AQ), psychological distress (K10), UADI-2, MPPUS, and alcohol

**Table 2. Dependent variables of the included participants (n = 420) by age groups.**

| Variables | Age groups | | | | | Statistical test | p-value |
| | Overall N = 420 | 18–24 n = 114 | 25–36 n = 107 | 37–49 n = 106 | 50–65 n = 93 | | |
|---|---|---|---|---|---|---|---|
| **AQ** | | | | | | H (3) = 18.14 | <0.001 |
| Mean (SD) | 16.84 (6.96) | 18.75 (7.68) | 17.80 (7.09) | 15.51 (6.72) | 14.91 (5.27) | | |
| Min - Max | 2 - 42 | 3 - 42 | 3 - 33 | 2 - 33 | 4 - 26 | | |
| **Autistic traits (AQ)** | | | | | | χ² (3, N= 420) = 21.21 | <0.001 |
| Low autistic traits | 359 (85%) | 88 (77%) | 85 (79%) | 96 (91%) | 90 (97%) | | |
| High autistic traits | 61 (15%) | 26 (23%) | 22 (21%) | 10 (9%) | 3 (3%) | | |
| **K10** | | | | | | H (3) = 39.38 | <0.001 |
| Mean (SD) | 22.32 (8.22) | 25.49 (7.61) | 23.12 (9.13) | 19.75 (7.08) | 20.42 (7.66) | | |
| Min - Max | 10 - 48 | 10 - 43 | 10 - 48 | 10 - 44 | 10 - 47 | | |
| **Psychological distress (K10)** | | | | | | χ² (9, N= 420) = 45.53 | <0.001 |
| No distress | 182 (44%) | 28 (25%) | 42 (40%) | 59 (57%) | 53 (58%) | | |
| Mild distress | 86 (21%) | 22 (19%) | 22 (21%) | 26 (25%) | 16 (18%) | | |
| Moderate distress | 69 (17%) | 31 (27%) | 20 (19%) | 9 (8.7%) | 9 (9.9%) | | |
| Severe distress | 77 (19%) | 32 (28%) | 22 (21%) | 10 (9.6%) | 13 (14%) | | |
| **UADI-2** | | | | | | H (3) = 88.90 | <0.001 |
| Mean (SD) | 56.99 (15.62) | 66.36 (13.56) | 59.94 (14.14) | 50.48 (13.92) | 49.52 (14.51) | | |
| Min - Max | 25 - 106 | 36 - 97 | 31 - 99 | 27 - 98 | 25 - 106 | | |
| **MPPUS** | | | | | | H (3) = 70.33 | <0.001 |
| Mean (SD) | 53.27 (16.00) | 61.41 (14.25) | 56.53 (14.71) | 47.80 (15.36) | 45.77 (14.57) | | |
| Min - Max | 24 - 96 | 24 - 96 | 26 - 95 | 24 - 93 | 24 - 85 | | |
| **ASSIST – Tobacco** | | | | | | H (3) = 29.29 | <0.001 |
| Mean (SD) | 6.95 (9.36) | 10.03 (10.35) | 8.50 (9.96) | 4.59 (7.64) | 4.09 (7.57) | | |
| Min - Max | 0 - 36 | 0 - 33 | 0 - 36 | 0 - 28 | 0 - 30 | | |
| **ASSIST – Alcohol** | | | | | | H (3) = 20.59 | <0.001 |
| Mean (SD) | 5.76 (5.58) | 7.17 (6.10) | 6.08 (5.13) | 5.13 (5.14) | 4.37 (5.55) | | |
| Min - Max | 0 - 27 | 0 - 24 | 0 - 23 | 0 - 22 | 0 - 27 | | |

AQ: Autistic Spectrum Quotient; K10: Kessler Psychological Distress Scale; UADI-2: Questionario Uso-Abuso e Dipendenza da Internet 2; ASSIST: The Alcohol, Smoking and Substance Involvement Screening Test

and tobacco use (ASSIST). The sample was mainly characterized by low autistic traits (85%, n = 359) with a mean score of 16.84 (SD = 6.96; 95% CI [16.17, 17.51]; range: 2 – 42). Additionally, the sample reported mild psychological distress (M = 22.32, SD = 8.22; 95% CI [21.65, 22.99]), normal Internet use (M = 56.99, SD = 15.62; 95% CI [55.69, 58.29]) and moderate smartphone use (M = 53.27, SD = 16; 95% CI [51.97, 54.57]). Younger participants (18 – 24 age-group) showed higher levels of Internet (M = 66.36, SD = 13.56; 95% CI [63.87, 68.85]) and mobile phone use (M = 61.41, SD = 14.25; 95% CI [58.92, 63.9]). Significant age groups differences were found in all outcomes: AQ (H (3) = 18.14, $p < .001$), autistic traits ($\chi^2$ (3, N = 420) = 21.21, $p < .001$), K10 (H (3) = 39.38, $p < .001$), psychological distress ($\chi^2$ (9, N = 420) = 45.53, $p < .001$), UADI-2 (H (3) = 88.90, $p < .001$), MPPUS (H (3) = 70.33 $p < .001$), ASSIST-tobacco (H (3) = 29.29, $p < .001$), and -alcohol use (H (3) = 20.59, $p < .001$).

Percentages and Chi-squared test results for the use of alcohol, tobacco, cannabis, cocaine, amphetamine, inhalants, sedatives, hallucinogens and opioids across age groups were reported in Supporting Information S9 Table.

## Correlational analysis

Table 3 shows the Spearman correlations between AQ and each dependent variable (UADI-2, MPPUS, ASSIST-alcohol and ASSIST-tobacco) across age groups, along with their 95% CI. AQ was weakly to moderately but significantly associated with UADI-2 in all age groups (ρ = 0. 22 – 0.50, all $p < .03$), while MPPUS was weakly and significantly associated in all age groups despite the 50 – 65 age-group (ρ = 0. 24 – 0.42, all $p < .01$). A weak negative but significant association was found between AQ and tobacco use (ρ = – 0.19, all $p = .049$) in young participants (18 – 24 age group). No significant association was found between AQ and alcohol use.

Table 4 presents the Spearman correlations between K10 and dependent variables (UADI-2, MPPUS, ASSIST-alcohol and ASSIST-tobacco) in each age group, with corresponding 95% CI. Results showed that K10 was significantly associated in all age groups with both UADI-2 (ρ = 0.28 – 0.59, all $p < .006$) and MPPUS (ρ = 0. 27 – 0.58, all $p < .009$).

**Table 3. Spearman correlations between AQ and dependent variables by age groups.**

| Age group | Dependent Variables | rho | 95% CI | p value | n |
|---|---|---|---|---|---|
| 18–24 | Alcohol (ASSIST) | -0.17 | [-0.34, 0.02] | 0.073 | 114 |
| 18–24 | Tobacco (ASSIST) | -0.19 | [-0.36, 0] | 0.049 | 114 |
| 18–24 | MPPUS | 0.41 | [0.24, 0.55] | <.001 | 114 |
| 18–24 | UADI-2 | 0.46 | [0.31, 0.6] | <.001 | 114 |
| 25–36 | Alcohol (ASSIST) | -0.04 | [-0.23, 0.15] | 0.691 | 107 |
| 25–36 | Tobacco (ASSIST) | -0.06 | [-0.24, 0.13] | 0.555 | 107 |
| 25–36 | MPPUS | 0.42 | [0.25, 0.56] | <.001 | 107 |
| 25–36 | UADI-2 | 0.50 | [0.35, 0.63] | <.001 | 107 |
| 37–49 | Alcohol (ASSIST) | 0.06 | [-0.13, 0.25] | 0.515 | 106 |
| 37–49 | Tobacco (ASSIST) | -0.09 | [-0.27, 0.1] | 0.369 | 106 |
| 37–49 | MPPUS | 0.24 | [0.05, 0.41] | 0.015 | 106 |
| 37–49 | UADI-2 | 0.30 | [0.12, 0.47] | 0.002 | 106 |
| 50–65 | Alcohol (ASSIST) | 0.14 | [-0.07, 0.33] | 0.194 | 93 |
| 50–65 | Tobacco (ASSIST) | 0.07 | [-0.14, 0.27] | 0.517 | 93 |
| 50–65 | MPPUS | 0.12 | [-0.08, 0.32] | 0.238 | 93 |
| 50–65 | UADI-2 | 0.22 | [0.02, 0.41] | 0.033 | 93 |

AQ: Autistic Spectrum Quotient; K10: Kessler Psychological Distress Scale; UADI-2:Uso-Abuso e Dipendenza da Internet 2; MPPUS: Mobile Phone Problem Usage Scale; ASSIST: The Alcohol, Smoking and Substance Involvement Screening Test

**Table 4. Spearman correlations between K10 and dependent variables by age groups.**

| Age group | Dependent Variable | rho | 95% CI | p value | n |
|---|---|---|---|---|---|
| 18–24 | Alcohol (ASSIST) | 0.19 | [0.01, 0.36] | 0.042 | 114 |
| 18–24 | Tobacco (ASSIST) | 0.12 | [-0.06, 0.3] | 0.189 | 114 |
| 18–24 | MPPUS | 0.49 | [0.33, 0.62] | <.001 | 114 |
| 18–24 | UADI-2 | 0.47 | [0.31, 0.6] | <.001 | 114 |
| 25–36 | Alcohol (ASSIST) | 0.28 | [0.09, 0.44] | 0.004 | 107 |
| 25–36 | Tobacco (ASSIST) | 0.28 | [0.09, 0.45] | 0.004 | 107 |
| 25–36 | MPPUS | 0.58 | [0.44, 0.7] | <.001 | 107 |
| 25–36 | UADI-2 | 0.59 | [0.46, 0.7] | <.001 | 107 |
| 37–49 | Alcohol (ASSIST) | 0.18 | [-0.01, 0.36] | 0.064 | 106 |
| 37–49 | Tobacco (ASSIST) | -0.03 | [-0.22, 0.16] | 0.762 | 106 |
| 37–49 | MPPUS | 0.46 | [0.3, 0.6] | <.001 | 106 |
| 37–49 | UADI-2 | 0.40 | [0.23, 0.55] | <.001 | 106 |
| 50–65 | Alcohol (ASSIST) | 0.18 | [-0.03, 0.37] | 0.091 | 93 |
| 50–65 | Tobacco (ASSIST) | 0.17 | [-0.03, 0.36] | 0.102 | 93 |
| 50–65 | MPPUS | 0.27 | [0.07, 0.45] | 0.009 | 93 |
| 50–65 | UADI-2 | 0.28 | [0.09, 0.46] | 0.006 | 93 |

AQ: Autistic Spectrum Quotient; K10: Kessler Psychological Distress Scale; UADI-2: Uso-Abuso e Dipendenza da Internet 2; MPPUS: Mobile Phone Problem Usage Scale; ASSIST: The Alcohol, Smoking and Substance Involvement Screening Test

Weak positive association between K10 and alcohol use in 18 – 24 ($\rho$ = 0.19, $p$ = .04) and 25 – 36 age-groups ($\rho$ = 0.28, $p$ = .004). Tobacco use was weakly and significantly associated with K10 only in the 25 – 36 age-group ($\rho$ = 0.28, $p$ = .004).

Spearman correlation results for the total sample are reported in Supporting Information S10 Table.

## MANCOVA and exploratory analysis

Assumptions were met for homogeneity of correlation matrices (Box's M test), homogeneity of univariate variances (Levene's test), and homogeneity of regression slopes, but not for multivariate normality distribution of the variables. After checking for outliers (Mahalanobis distance), the MANCOVA can be performed using the Pillai's trace as more robust test when the normality is violated. Moreover, multicollinearity was assessed by examining Spearman's correlations among the dependent variables (UADI-2, MPPUS, ASSIST-alcohol and ASSIST-tobacco), and by computing variance inflation factors (VIFs). Spearman correlations were low to moderate ($\rho$ = 0.16 – 0.34) between substance use measures and Internet-related behaviours. However, UADI-2 and MPPUS were strongly correlated ($\rho$ = 0. 79), indicating an overlap between these measures. All VIFs were all below the threshold of 5 [72] (range = 1.03 – 2.65), indicating that multicollinearity was not sufficiently severe to compromise the stability of the MANCOVA. On this basis, and given the robustness of Pillai's trace, all dependent variables were included in the MANCOVA.

MANCOVA assess if autistic traits (low vs high autistic traits) and the covariate (K10) predicted the combined dependent variables (UADI-2, MPPUS, ASSIST-alcohol and ASSIST-tobacco). Results showed a significant effect of both autistic traits (V = 0.11, F(4, 417) = 13.41, $p$ < .001, $\eta^2$ partial = .11; 95% CI [0.07, 1.00]) and psychological distress (V = 0.28, F(4, 417) = 41.97, $p$ < .001, $\eta^2$ partial = .29; 95% CI [0.23, 1.00]) in the prediction of the combined variables.

Follow-up ANCOVAs assessed factors (autistic traits, sociodemographic variables and psychological distress) for each dependent variable:

- Alcohol use (ASSIST): significant factors included K10 (F(1,362) = 23.38, *p* < .001, η² = .06; 95% CI [0.03, 1.00]), age groups (F(3,362) = 3.72, *p* = 0.01, η² = .03) and occupation (F(5,362) = 2.69, *p* = 0.02, η² = .04). Post-hoc comparisons with Tukey's tests did not reveal any significant differences in age groups or occupation.

- Tobacco use (ASSIST): significant factors were K10 (F(1, 362) = 39.37, *p* < .001, η² = .10; 95% CI [0.05, 1.00]), age groups (F(3, 362) = 8.59, *p* < .001, η² = .07), marital status (F(5,362) = 2.38, *p* = .03, η² = .03), type of mobile phone use (F(5, 362) = 2.27, *p* = .04, η² = .03), and familiarity (F(2, 362) = 3.15, *p* = .04, η² = .02). Tukey's tests revealed no pairwise differences in significant factors.

- PIU (UADI-2): significant factors were autistic traits (F(1, 362) = 61.66, *p* < .001, η² = .15; 95% CI [0.09, 1.00]), K10 (F(1, 362) = 152.26, *p* < .001, η² = .30; 95% CI [0.23, 1.00]), age groups (F(3, 362) = 21.98, *p* < .001, η² = .15), time spent on mobile phone (F(3, 362) = 10.77, *p* < .001, η² = .08), time spent on social networks (F(4, 362) = 4.66, *p* = .001, η² = .05), and issues with justice (F(2, 362) = 3.07, *p* = .04, η² = .02). Tukey's tests showed significant pairwise differences in time spent on mobile phone, on social network and issues with justice (see Supporting Information S11 Table).

- PMPU (MPPUS): autistic traits (F(1, 362) = 33.25, *p* < .001, η² = .08; 95% CI [0.04, 1.00]), K10 (F(1, 362) = 157.22, *p* < .001, η² = .30; 95% CI [0.024, 1.00]), age groups (F(3, 362) = 15.68, *p* < .001, η² = .12), type of mobile phone use (F(3, 362) = 20.25, *p* < .001, η² = .14), time spent on social networks (F(4, 362) = 4.35, *p* = .001, η² = .05) and issues with justice (F(2, 362) = 4.03, *p* = .01, η² = .02) showed significant effects. Tukey's tests revealed significant pairwise differences in time spent on mobile phone, on social network and issue with justice (see Supporting Information S11 Table), while no pairwise comparisons revealed significant differences between age groups.

In summary, autistic traits showed significant effects on PIU and PMPU, though not on alcohol and tobacco use. Psychological distress (K10) and age groups showed consistent effects across all outcomes, while several sociodemographic variables contributed variably. Details of all ANCOVAs and of Tukey's tests are reported in the Supporting Information S11 Table.

Finally, the sensitivity MANCOVA including age as a continuous covariate showed that age (V = 0.18, F(4, 413) = 23.11, *p* < .001), autistic traits (V = 0.12, F(4, 413) = 14.62, *p* < .001) and K10 (V = 0.33, F(4, 413) = 50.91, *p* < .001) had a significant effect in the prediction of combined dependent variables. These results confirmed that the association between autistic traits and the outcomes remain robust and statistically significant when controlling for age as continuous variables and regardless the deviation from pre-registration.

### Hurdle models

The zero Hurdle Model results indicated that autistic traits did not impact the probability of becoming a user of tobacco (*β* = 0.41, p = .16) or alcohol (*β* = 0.20, p = .58). In contrast, psychological distress remained a highly significant predictor both of tobacco (*β* = 0.05, p < .001) and alcohol use (*β* = 0.06, p < .001). The count model further showed that, among tobacco users, low autistic traits significantly predicted the intensity of tobacco use (*β* = 0.31, p = .03). Specifically, among tobacco users, those with lower autistic traits showed higher tobacco use, compared to those with high autistic traits. Instead, for alcohol users, low autistic traits were not a significant predictor of alcohol use (*β* = 0.20, p = .07). Psychological distress (K10) was significantly associated with higher levels of use among both tobacco and alcohol users (*β* = 0.02, p < .001 for both substances). These findings further support the pre-registered MANCOVA results, with psychological distress showing a consistent association with substance use, while the effect of autistic traits appear more limited on them. Details of Hurdle Models were reported in Supporting Information S12 Table.

### Multivariate Multiple Regression (MMR)

MMR models produced results consistent with previous analyses. PIU (UADI-2) was significantly predicted by both AQ (β = 0.59, SE = 0.10, 95% CI[0.40, 0.79], *t*(417) = 5.93, *p* < .001) and K10 (β = 0.74, SE = 0.08, 95% CI [0.57, 0.91], *t*(417)

= 8.68, *p*<.001). These resul*ts* indicate that higher level of autistic traits and psychological distress were both associated with higher level of PIU.

PMPU (MPPUS) was also significantly predicted by both AQ (β = 0.37, SE = 0.10, 95% CI[0.17, 0.58], *t*(417) = 3.54, *p*<.001) and K10 (β = 0.84, SE = 0.08, 95% CI[0.66, 1.02], *t*(417) = 9.37, *p*<.001), indica*ti*ng that higher autistic traits and psychological distress were both related to higher level of PMPU.

Alcohol use (ASSIST) was significantly predicted by K10 (β = 0.15, SE = 0.03, 95% CI[0.09, 0.23], t(417) = 4.38, *p*<.001), but not by AQ (β = –0.03, SE = 0.04, 95% CI[-0.12, 0.04], *t*(417) = –0.89, *p* = .373). Thus, higher psychological dis*t*ress was associated with higher alcohol consumption.

Tobacco use was significantly predicted by both K10 (β = 0.36, SE = 0.05, 95% CI[0.25, 0.48], t(417) = 6.14, *p*<.001) and AQ (β = –0.18, SE = 0.06, 95% CI[-0.32, -0.04], *t*(417) = –2.59, *p* = .009). Higher psychological dis*t*ress was associated with greater tobacco use, whereas higher autistic traits were associated with lower tobacco use.

According to MMR, all outcomes were significantly predicted by psychological distress (K10), while autistic traits (AQ) specifically predicted PIU (UADI-2) and PMPU (MMPUS). In both cases, higher level of psychological distress and autistic traits were associated with greater problematic use, expect for tobacco use. The only divergence between MMR and MANCOVA results was that AQ additionally predicted tobacco use in MMR model.

## Discussion

The first research question of the present study examined whether the frequency of PIU, PMPU, psychological distress and substances use varied across age groups. Across all outcomes, younger participants showed higher level of problematic behaviours and psychological distress, which tended to decrease progressively with age. In the overall sample, the Internet use was normal (M = 56.99, SD = 15.62) and the use of mobile phone was moderate (M = 53.27, SD = 16), but these levels were higher among younger participants aged 18 – 24 years old (PIU: M = 66.36, SD = 13.56; PMPU: M = 61.41, SD = 14.25). Participants aged 25 – 36 years showed similar, though slightly lower, levels of PIU and PMPU. The higher level of PIU and PMPU among young adults aligns with previous evidences indicating that younger populations are particularly vulnerable to Internet-related behaviours [11,73–75]. Participants aged up to 36 years reported using their mobile phone mainly to communicate (e.g., messages, calls, emails) and to attend social network, with 47% of those aged 18 – 24 reporting more than 7 hours of daily social media use. In contrast, older participants primarily use their phones for communication purpose. This finding is consistent with a recent study which shows that Generation X and Generation Y (Millennials) use their phones in different ways, with Millennials more likely to engage in asynchronous communication (e.g., social media), and Generation X preferring synchronous communication (e.g., calls) [76].

Regarding psychological distress, while 44% (n = 182) of the total sample reported no symptoms, a substantial proportion of young adults (18 – 24 years) showed severe distress (28%, n = 32). This pattern is consistent with numerous studies highlighting elevated levels of psychological distress among young adults and university students worldwide [77–79].

A similar age-related pattern emerged for substance use. Alcohol consumption was common across all age groups but more frequent among younger participants (below 36 years), who also reported higher rated of tobacco use in contrast to older participants who rarely used tobacco. These results are in line with previous research showing that substance use generally decrease with age [80,81]. Considering that younger participants in this study also exhibited higher psychological distress, it is plausible that such distress contributes to increased engagement in risky behaviours, including substance misuse.

The second research question investigated whether different levels of autistic traits were associated with differences in PIU, PMPU and substance use, after controlling for psychological distress. Our findings consistently indicate that higher autistic traits and greater level of psychological distress are both associated with increased levels of PIU and PMPU. As reported earlier, the association between autistic traits and PIU/PMPU can be explain by two core features of autism spectrum: (a) deficit in social interactions and communication, and (b) RIRBs (e.g., gaming) [31,50]. According to the social

compensation hypothesis, Internet may offer a valid alternative to face-to-face interactions, providing a less demanding social environment for individual with social-communicative challenges, like those with high autistic traits. Furthermore, individuals with high autistic traits may be particularly drawn to non-social use of Internet, engaging in information-seeking behaviours that support their RIRBs [82]. For instance, gaming platforms can cater to RIRBs through immersive and customizable digital environments, where users can also control sensory inputs (e.g., sounds, visual features). Shane-Simpson and colleagues (2016) further argued that the severity of RRIBs may contributed more strongly to PIU in both individuals with ASD and those with subclinical traits [50].

Nevertheless, the correlational analysis suggested that psychological distress generally showed a stronger association with PIU and PMPU than autistic traits, particularly among participants aged 25 – 36. Psychological distress consistently showed stronger associations with PIU and PMPU than autistic traits across ANCOVAs, MANCOVA and MMR model, as reflected by larger standardized coefficient and higher statistical significance. This aligns with extensive evidence highlighting psychological distress as key factor associated with PIU [83–87]. For instance, a cross-sectional survey of 449 Australians found that PIU was associated with higher level of psychological distress (particularly anxiety), and avoidance coping mediated this relationship [88]. Similarly, a meta-analytic review reported that PIU was positively associated with depressive and anxiety symptoms, loneliness, and negatively associated with well-being [89].

Our finding also indicates that psychological distress is consistently associated with tobacco and alcohol use, while autistic traits showed no robust association with these behaviours. The relationship between substance use and psychological distress is well established in literature [90–92], whereas the association between substance use and autistic traits is less clear and has been less studied [47,48]. Factors contributing to the development of SUD in ASD population have been linked to social environment and personality traits, such as limited social resources, reduced influence of peers, and low sensation-seeking [46]. Moreover, the risk for SUD to increase in ASD population in comorbidity with ADHD, externalizing behaviour and in presence of family history of SUD [93–95]. In the current study, among participants with high autistic traits (n = 61), only one reported an ADHD diagnosis, 11 reported familiarity for SUD and 5 for behavioural disorders. Therefore, our high-autistic-traits participants may not represent a population at elevated risk for SUD, which could partially explain the lack of association between autistic traits and substance use. Nonetheless, it should also be noted that MMR results showed that higher autistic traits predicted lower tobacco use, and correlational analysis indicated a significant negative association between autistic traits and tobacco use among participants aged 18 – 24 years. Hurdle Model results additionally specified that, among tobacco users, those with lower autistic traits showed higher tobacco use. Therefore, these findings are consistent with previous evidence suggesting that individuals with higher autistic traits may exhibit lower levels of substance use. However, the link between autistic traits and substance use, in particular tobacco, should be more deeply investigated to understand the underlying mechanisms driving this relationship.

Although the cross-sectional design of this study precludes causal inferences, these findings suggest that psychological distress can be broadly associated with PIU, PMPU and substance use, whereas autistic traits may be more specifically related to PIU and PMPU. In other words, autistic traits may represent a potential factor contributing to vulnerability to PIU and PMPU, but not to substance abuse. This distinction may indicate that PIU/PMPU and substance use may follow different pathways, in line with scholars who have criticized the classification of PIU and PMPU as addiction and have emphasized the need to reconceptualize these behaviours [96–98]. For instance, Billieux and colleagues (2015) [97] argued that applying traditional criteria for behavioural addiction almost any excessive involvement in any kind of activity can be labelled as psychiatric disorder, leading to an overpathologization of daily behaviours. This categorical, simplistic and symptom-based approach risks to oversimplify complex and heterogeneous behaviours, hindering the identification of specific mechanisms and the development of tailored intervention. Regarding PMPU, Billieux and colleagues further highlighted the lack of neurobiological and behavioural (e.g., tolerance) evidence supporting its classification as an addiction, and instead proposed a theoretical model composed by distinct pathways – reassurance, impulsive and extraversion

PLOS Mental Health

pathway – which can drive different aspects of PMPU [99]. Nevertheless, these interpretations should be considered with caution in light of study's limitations (see next paragraph).

Overall, these findings partially align with literature reporting elevated PIU in individuals with high autistic traits [31,32,36–38], but also suggest that autistic traits alone do not fully explain PIU. The association between autistic traits, psychological distress i and PIU and PMPU may be understood within the I-PACE model, whereby autistic traits may function as one of the potential predisposing factors, and psychological distress may act as an affective factor influencing the relationship with Internet-related behaviours. Within this framework, psychological distress may encourage individuals with elevated autistic traits to use the Internet as an emotion-regulation strategy. Environmental influences may further encourage this patter: for instance, parents may encourage Internet use to manage overstimulation or distress of their children with ASD [100,101]. Moreover, it is possible that individuals with high autistic traits, especially during adolescence, may be particularly vulnerable to Internet misuse, considering that their use of social networks often differs from that of their neurotypical peers. Specifically, they may tend to prefer either game's chats instead of classical social networks or other less intense social interactions [31,102]. Such patterns may foster isolation, reinforce RIRBs and increase Internet-related disorders risk, along with exposure to cyberbullying that perpetuates a maladaptive feedback loop [102]. Parents thus play a crucial role in the regulation of contents, frequency, and duration of online activities [102]. A recent systematic review by Nielsen and colleagues (2020) found that positive parenting and positive dynamics were associated with lower PIU, while negative parenting and dynamics increased its risk [103]. Similarly, another study has shown that negative parenting style and stress explain a significant proportion of PIU in children with ASD [104].

## Limitations and future studies

This study has several limitations. First, the cross-sectional design, although common in the field, limits generalizability and the causal interpretation of the results. Second, despite the UADI-2 has good psychometric properties, it is standardized only on Italian sample. This implies that our findings may not be applicable to other cultural or geographical settings. Moreover, UADI-2 and MPPUS partially overlap, and there is ongoing debate on whether Internet and smartphone overuse should be considered as distinct constructs. Third, self-report measures may underestimate substance intake, considering illegality of many of them. Fourth, the tertile-based categorization does not provide a true estimate of PMPU prevalence, as it is sample-dependent. Although it was employed solely for descriptive purposes, it is not standardized and therefore limits the comparability with other sample. Fifth, the non-representative nature of the sample reduces the generalizability of findings. Specifically, the sample was predominantly female (79%, n = 331), and mostly from Italian islands (63%, n = 262). Living in insular contexts may increase the perception of social isolation and loneliness, factors known to heighten PIU risk, particularly among adolescents and young adults [105–107]. These demographic characteristics should be carefully considered when interpreting the findings. Sixth, although the overall sample did not show particularly high levels of PIU or PMPU, recruitment via snowball sampling through social networks and the online administration of the survey may have preferentially selected individuals who use digital devices and Internet more frequently. This may have partially biased the results by overrepresenting, for example, digitally competent or socially isolated individuals, which could lead to greater time spent on Internet. Nevertheless, to the best of our knowledge, this is the first study that examined the interplay between autistic traits, psychological distress, PIU, PMPU and substances use in the Italian population.

Future research should employ Ecological Momentary Assessment (EMA) and longitudinal design to capture real-time emotional state and clarify the temporal development of PIU and PMPU, and their link with psychological distress. Given the observed association between psychological distress and Internet-related behaviours and in line with the I-PACE model, interventions should prioritize the reduction of psychological distress as affective factor influencing coping skills and emotion-regulation strategies, as well as address family environment dynamics that may contribute to PIU and PMPU. Cognitive-behavioural interventions aimed at improving coping skills and emotion-regulation capacities may be particularly

beneficial, as problematic Internet and mobile phone use can function as avoidant or compensatory strategy for managing negative affect from early stages. Parenting-focused psychoeducational intervention (e.g., guidance on monitoring screen time during sensitive developmental periods), or the inclusion of caregiver modules within adolescent interventions, may be especially relevant to impact on environmental reinforcement processes. Interventions should also consider the assessment of elevated autistic traits or high level of RIRBs, particularly among adolescents, to address a potential predisposing factor (i.e., autistic traits) underlying Internet-related behaviours.

## Conclusions

Our findings confirm that Internet use plays an increasingly central role in everyday life across age groups, with multiple factors influencing its potential misuse, including autistic traits and psychological distress. Younger adults (18 – 24 years) consistently reported higher psychological distress, greater Internet and mobile phone misuse, and more frequent substance use compared to older age groups. These associations highlight the importance of prevention and intervention strategies that focus on reducing psychological distress and enhancing adaptive emotion-regulation skills, while also screening for elevated autistic traits. Future efforts should aim to develop developmentally sensitive interventions targeting Internet-related behaviours that also involve parents. University mental health services could contribute by providing psychoeducation on the risks of Internet and smartphone overuse, offering coping-skills and emotional-regulation interventions, and implementing targeted prevention programs.

## Supporting information

**S1 Table. Differences between participants who completed the IAT and those who did not.**
(DOCX)

**S2 Table. Main analyses excluding the Imagination subscale of the AQ.**
(DOCX)

**S3 Table. Description of sociodemographic variables.**
(DOCX)

**S4 Table. Descriptive analysis of subsample of males and females (n = 414) by gender.**
(DOCX)

**S5 Table. MANOVA and ANCOVAs of subsample (n = 414, males and females).**
(DOCX)

**S6 Table. Other sociodemographic variables among included participants (n = 420).**
(DOCX)

**S7 Table. Psychological diagnosis details.**
(DOCX)

**S8 Table. Use of mobile phone and Internet among participants by age groups.**
(DOCX)

**S9 Table. Percentages and Chi-squared of substances use.**
(DOCX)

**S10 Table. Spearman correlation between AQ or K10 and dependent variables among all participants.**
(DOCX)

**S11 Table. ANCOVAs and Tukey's tests.**
(DOCX)

**S12 Table. Hurdle Models.**
(DOCX)

## Acknowledgments

Authors thanks Prof. Silvia Casale for her valuable suggestions and guidance during project management phase and the manuscript preparation.

## Author contributions

**Conceptualization:** Matilda Floris, Claudio Gentili.

**Data curation:** Matilda Floris.

**Formal analysis:** Matilda Floris.

**Investigation:** Matilda Floris.

**Methodology:** Matilda Floris.

**Project administration:** Claudio Gentili.

**Supervision:** Claudio Gentili.

**Writing – original draft:** Matilda Floris, Claudio Gentili.

**Writing – review & editing:** Matilda Floris, Claudio Gentili.

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
