## [Decision Letter · Decision Letter 0]

11 Jan 2026

PMEN-D-25-00468

Association Between Problematic Internet and Mobile Phone Use, Autistic Traits, and Psychological Distress Among Adults: A Cross-Sectional Survey

PLOS Mental Health

Dear Dr. Floris,

Thank you for submitting your manuscript to PLOS Mental Health. After careful consideration, we feel that it has merit but does not fully meet PLOS Mental Health’s publication criteria as it currently stands. Therefore, we invite you to submit a revised version of the manuscript that addresses the points raised during the review process.

We look forward to receiving your revised manuscript.

Kind regards,

Giuseppe Carrà, PhD

Academic Editor

PLOS Mental Health

Journal Requirements:

i. Please clarify all sources of financial support for your study. List the grants, grant numbers, and organizations that funded your study, including funding received from your institution. Please note that suppliers of material support, including research materials, should be recognized in the Acknowledgements section rather than in the Financial Disclosure.

ii. State the initials, alongside each funding source, of each author to receive each grant. For example: "This work was supported by the National Institutes of Health (####### to AM; ###### to CJ) and the National Science Foundation (###### to AM)."

iii. State what role the funders took in the study. If the funders had no role in your study, please state: “The funders had no role in study design, data collection and analysis, decision to publish, or preparation of the manuscript.”

iv. If any authors received a salary from any of your funders, please state which authors and which funders.

2. Please provide separate figure files in .tif or .eps format.

https://journals.plos.org/mentalhealth/s/figures

https://journals.plos.org/mentalhealth/s/figures#loc-file-requirements

4. In the online submission form, you indicated that “The collected data included sensitive informations (e.g., IP addresses, response IDs, medical and psychological diagnosis, traumatic events, justice-related experiences). Therefore, the dataset is available from the corresponding author upon motivated request.”.

3. Uploaded as supplementary information.

Additional Editor Comments (if provided):

Reviewers' comments:

Reviewer's Responses to Questions

**Comments to the Author**

1. Does this manuscript meet PLOS Mental Health’s publication criteria? Is the manuscript technically sound, and do the data support the conclusions? The manuscript must describe methodologically and ethically rigorous research with conclusions that are appropriately drawn based on the data presented.

Reviewer #1: Yes

Reviewer #2: Yes

2. Has the statistical analysis been performed appropriately and rigorously?

Reviewer #1: Yes

Reviewer #2: Yes

3. Have the authors made all data underlying the findings in their manuscript fully available (please refer to the Data Availability Statement at the start of the manuscript PDF file)?

Reviewer #1: No

Reviewer #2: Yes

4. Is the manuscript presented in an intelligible fashion and written in standard English?

Reviewer #1: Yes

Reviewer #2: Yes

5. Review Comments to the Author

Reviewer #1: The manuscript describes the analysis of a cross-sectional survey on Italian adults (n=420), examining associations between autistic traits, problematic internet/mobile phone use (PIU/PMPU), psychological distress, and substance use. The results show that autistic traits correlated with PIU/PMPU, and psychological distress correlates with all outcomes. The manuscript is clearly written, and the topic is relevant for PLOS Mental Health. However, there are some methodological and reporting issues that should be addressed. I detail them below.

Preregistration and data availability:

The link to the preregistration does not work for me, so it is impossible to check: https://osf.io/d2nbg

We know from substantial literature now that datasets "available from the corresponding author upon motivated request" are often difficult, if not impossible, to access.

The authors are required to make available online the preregistration, as well as an anonymised version of the dataset and the codes used to analyse it.

Generalisation and causal language in the discussion:

The authors recognise some of the limitations e.g., that the sample is heavily skewed (80% female, 63% from Italian islands) and that "the cross-sectional design [...] limits generalizability and the causal interpretation of the results". However, the discussion moves toward causal language (e.g., “autistic traits may increase the vulnerability to develop PIU and PMPU”) and broadly generalises.

Related, convenience and especially snowball recruitment via social media may also be problematic. Given the topic (digital use), recruitment channels may have selected individuals already more online, more distressed, or more neurodivergent. This also requires more explicit acknowledgement.

Variables validity/Analysis:

The authors do a good job in explaining and justifying their variable choices. Here are a few concerns:

The AQ sub-domains have "poor to acceptable internal consistency". E.g., the imagination subscale has α = .37. Would this instability propagate into the main predictor? Maybe an analysis with only the dimensions that show acceptable internal consistency could be presented to reassure about the robustness of the results in this respect?

Given the absence of data, it is impossible to check, but it is plausible that ASSIST alcohol and ASSIST tobacco are zero-inflated variables. Even if the authors use non-parametric tests, they assume standard continuous values, which can create a problem.

Psychological distress, PIU, and PMPU are all moderately to strongly interrelated. Using them simultaneously in MANCOVA without addressing multicollinearity risks unstable estimates.

Minor points:

Double-check the references, e.g. "Elhai JD, Dvorak RD, Levine JC, Hall BJ. Problematic smartphone use..." is reported twice.

Several tables and statistical results are reported in the manuscript, making it difficult to read. I would suggest selecting what the important ones are, and moving the rest to supplementary material or similar.

Reviewer #2: In this manuscript, the authors present an online cross-sectional survey (N=420) examining associations between autistic traits (AQ), problematic Internet use (UADI-2), problematic mobile phone use (MPPUS), psychological distress (K10), substance use (ASSIST), and age in an Italian community sample. Higher autistic traits and—more consistently—higher psychological distress are associated with greater PIU/PMPU, with younger adults showing the highest levels.

The topic is timely and the manuscript is generally clear and well structured. However, several points should be clarified to improve reproducibility and interpretation.

Introduction

• The authors should explicitly state how PIU and PMPU were defined and measured.

• The authors should briefly introduce the I-PACE model (if used to guide interpretation) already in the Introduction, not only in the Discussion.

• The authors should clarify that the focus is on autistic traits in the general population, not clinical ASD diagnosis.

Methods

• The authors should briefly discuss likely selection effects from convenience/social-media recruitment and implications for prevalence estimates.

• The authors should more clearly report the deviation from preregistration regarding age grouping, and consider a sensitivity analysis using age as a continuous predictor/covariate.

• The authors should clarify outcome classifications:

o For UADI-2, the authors should specify whether “PIU prevalence” refers to misuse only or misuse + addiction, as current wording appears inconsistent with the categorical reporting.

o For MPPUS, the authors should justify tertile-based “high use” as a proxy for PMPU prevalence and emphasize continuous analyses as primary, given sample-dependence.

• The authors should clarify why only a subsample completed the IAT, and whether this subsample differs from the full sample.

• The authors should report basic multicollinearity checks (given conceptual overlap between PIU and PMPU) and, if feasible, include confidence intervals for key coefficients.

Implications

• The authors should more tightly connect intervention implications to the observed pattern (distress strongly linked to PIU/PMPU), highlighting distress-focused, CBT-informed coping/emotion-regulation strategies, with autism-adapted delivery when relevant.

• The authors should consider specifying whether a fully de-identified dataset + codebook could be shared to support reproducibility while respecting ethical constraints.

Minor edits: The authors should perform a brief language/typo pass (e.g., “cut-off”, “back button”, “imprisoned”).

6. PLOS authors have the option to publish the peer review history of their article (what does this mean?). If published, this will include your full peer review and any attached files.

**Do you want your identity to be public for this peer review?** For information about this choice, including consent withdrawal, please see our Privacy Policy.

Reviewer #1: No

Reviewer #2: No

Figure Resubmissions:

---

## [Decision Letter · Decision Letter 1]

8 Mar 2026

PMEN-D-25-00468R1

Association Between Problematic Internet and Mobile Phone Use, Autistic Traits, and Psychological Distress Among Adults: A Cross-Sectional Survey

PLOS Mental Health

Dear Dr. Floris,

Thank you for submitting your manuscript to PLOS Mental Health. After careful consideration, we feel that it has merit but does not fully meet PLOS Mental Health’s publication criteria as it currently stands. Therefore, we invite you to submit a revised version of the manuscript that addresses the points raised during the review process.

We look forward to receiving your revised manuscript.

Kind regards,

Giuseppe Carrà, PhD

Academic Editor

PLOS Mental Health

**Journal Requirements:**

Please review your reference list to ensure that it is complete and correct. If you have cited papers that have been retracted, please include the rationale for doing so in the manuscript text, or remove these references and replace them with relevant current references. Any changes to the reference list should be mentioned in the rebuttal letter that accompanies your revised manuscript. If you need to cite a retracted article, indicate the article’s retracted status in the References list and also include a citation and full reference for the retraction

notice.

**Additional Editor Comments (if provided):**

Reviewers' comments:

Reviewer's Responses to Questions

**Comments to the Author**

1. If the authors have adequately addressed your comments raised in a previous round of review and you feel that this manuscript is now acceptable for publication, you may indicate that here to bypass the “Comments to the Author” section, enter your conflict of interest statement in the “Confidential to Editor” section, and submit your "Accept" recommendation.

Reviewer #1: All comments have been addressed

Reviewer #2: All comments have been addressed

2. Does this manuscript meet PLOS Mental Health’s publication criteria? Is the manuscript technically sound, and do the data support the conclusions? The manuscript must describe methodologically and ethically rigorous research with conclusions that are appropriately drawn based on the data presented.

Reviewer #1: Yes

Reviewer #2: Yes

3. Has the statistical analysis been performed appropriately and rigorously?

Reviewer #1: Yes

Reviewer #2: Yes

4. Have the authors made all data underlying the findings in their manuscript fully available (please refer to the Data Availability Statement at the start of the manuscript PDF file)?

Reviewer #1: Yes

Reviewer #2: Yes

5. Is the manuscript presented in an intelligible fashion and written in standard English?

Reviewer #1: Yes

Reviewer #2: Yes

6. Review Comments to the Author

Reviewer #1: The authors have responded constructively to all my comments, thank you. The changes are well implemented. The only remaining concern is about the AQ imagination subscale sensitivity analysis, which was performed but not included in the manuscript or supplementary materials. The authors offered to do it if required. I would leave the decision to the editor. Otherwise, the manuscript is for me publishable in this form.

Reviewer #2: The authors have addressed most of the reviewer’s major concerns and the revised manuscript is substantially improved, particularly in clarifying the definitions and measurement of PIU/PMPU, introducing the I-PACE framework earlier, strengthening the distinction between autistic traits and clinical ASD, and adding a sensitivity analysis for age. However, a few issues remain only partially resolved in the manuscript itself, including: (i) clearer reporting of the IAT subsample, (ii) fuller presentation of confidence intervals for key results, (iii) a more explicit statement on data/code/codebook availability, and (iv) a final language edit. As a whole, the revision is largely responsive, and the remaining issues appear addressable without major additional analyses.

7. PLOS authors have the option to publish the peer review history of their article (what does this mean?). If published, this will include your full peer review and any attached files.

**Do you want your identity to be public for this peer review?** For information about this choice, including consent withdrawal, please see our Privacy Policy.

Reviewer #1: No

Reviewer #2: No

Figure Resubmissions:

---

## [Decision Letter · Decision Letter 2]

12 Apr 2026

Association Between Problematic Internet and Mobile Phone Use, Autistic Traits, and Psychological Distress Among Adults: A Cross-Sectional Survey

PMEN-D-25-00468R2

Dear Ms Floris,

We are pleased to inform you that your manuscript 'Association Between Problematic Internet and Mobile Phone Use, Autistic Traits, and Psychological Distress Among Adults: A Cross-Sectional Survey' has been provisionally accepted for publication in PLOS Mental Health.

Best regards,

Giuseppe Carrà, PhD

Academic Editor

PLOS Mental Health

Reviewer Comments (if any, and for reference):

Reviewer's Responses to Questions

**Comments to the Author**

1. If the authors have adequately addressed your comments raised in a previous round of review and you feel that this manuscript is now acceptable for publication, you may indicate that here to bypass the “Comments to the Author” section, enter your conflict of interest statement in the “Confidential to Editor” section, and submit your "Accept" recommendation.

Reviewer #2: All comments have been addressed

2. Does this manuscript meet PLOS Mental Health’s publication criteria? Is the manuscript technically sound, and do the data support the conclusions? The manuscript must describe methodologically and ethically rigorous research with conclusions that are appropriately drawn based on the data presented.

Reviewer #2: Yes

3. Has the statistical analysis been performed appropriately and rigorously?

Reviewer #2: Yes

4. Have the authors made all data underlying the findings in their manuscript fully available (please refer to the Data Availability Statement at the start of the manuscript PDF file)?

Reviewer #2: Yes

5. Is the manuscript presented in an intelligible fashion and written in standard English?

Reviewer #2: Yes

6. Review Comments to the Author

Reviewer #2: The authors have addressed the reviewers’ comments appropriately, and the manuscript has been revised accordingly.

7. PLOS authors have the option to publish the peer review history of their article (what does this mean?). If published, this will include your full peer review and any attached files.

**Do you want your identity to be public for this peer review?** For information about this choice, including consent withdrawal, please see our Privacy Policy.

Reviewer #2: No
